# Identification of Cultivated Land Quality Grade Using Fused Multi-Source Data and Multi-Temporal Crop Remote Sensing Information

Yinshuai Li [1], Chunyan Chang [1], Zhuoran Wang [1], Tao Li [2], Jianwei Li [2] and Gengxing Zhao [1,*]

[1] National Engineering Laboratory for Efficient Utilization of Soil and Fertilizer Resources, College of Resources and Environment, Shandong Agricultural University, Tai'an 271018, China; 2020110360@sdau.edu.cn (Y.L.); chyan0103@sdau.edu.cn (C.C.); wangzr@sdau.edu.cn (Z.W.)

[2] Soil & Fertilizer Working Station of Shandong Province, Jinan 250013, China; yushunzhang@shandong.cn (T.L.); lijianwei@shandong.cn (J.L.)

* Correspondence: zhaogx@sdau.edu.cn

**Abstract:** To explore the fast, accurate, and efficient remote sensing identification method of cultivated land quality, this study took Shandong Province as the study area, and used measured data to carry out the soil quality evaluation based on conventional GIS. On this basis, MODIS sequence images were used as remote sensing data sources, and multi-source data such as topography, meteorology, and statistical yearbook were fused. Then, according to the Pressure-State-Response framework, we constructed three kinds of characteristic indicators through distinguishing crop rotation types and fusing remote sensing data. Finally, the soil quality grade was identified by the random forest method, and the accuracy analysis was carried out. The results showed that the *NDVI* peak values of double-season crops are in mid-April and mid-August, and one-season crops are in mid-August. Through evaluation, soil quality was divided into three categories, with six grades. Through principal component analysis, each soil status indicator contains two to three principal components, and each principal component contains five to eight temporal crop remote sensing information. After distinguishing crop rotation types and fusing remote sensing images, the identification accuracy of soil quality is significantly improved. The overall accuracy is 79.18%, 86.12%, and 93.65%, and the Kappa coefficient is 0.66, 0.77, and 0.90, respectively. This research developed an automatic identification method for cultivated land quality grade, and it proved that distinguishing crop rotation types and fusing multi-temporal crop remote sensing information are effective ways to improve identification accuracy.

**Keywords:** soil quality; crop remote sensing; random forest; principal component analysis; Shandong Province; China

## 1. Introduction

Cultivated land is the material basis for agricultural production [1]. Soil quality is related to the ability of the cultivated land system to maintain biological productivity, protect ecological environment quality and promote animal and plant health. It is essential to ensure food security, protect biodiversity and maintain sustainable socio-economic development [2,3]. At present, the quantity decline and quality degradation of cultivated land resources have become a global problem. China is a country with a large population. It has become the foundation of sustainable agricultural development to realize the sustainable intensification of cultivated land and improve the soil quality [4,5]. Therefore, it is urgent for agricultural production and sustainable development to scientifically evaluate soil quality and give full play to the potential of cultivated land production.

Soil quality is a complex composed of physical, chemical, and biological environmental characteristics, and a single soil attribute cannot directly reflect the overall soil quality.

Its evaluation must consider the inherent properties, dynamic changes, soil processes, and the interaction with the external environment. At the same time, it is required to quickly and accurately identify the main limiting factors of crop production. Therefore, soil quality evaluation is one of the research focuses of soil science [6,7]. At present, soil quality evaluation has gradually developed from qualitative [8] to quantitative, and the soil quality index method is the most widely used quantitative evaluation method. For example, Andrews et al. [9], Xue et al. [10], and Nabiollahi et al. [11] combined multiple soil properties into a comprehensive soil quality index at different scales and regions. However, this method requires many field investigations and laboratory tests, and the repeated observation period is long. Thus, it is difficult to achieve rapid evaluation and dynamic monitoring of soil quality. How to simplify the evaluation process and realize the rapid and accurate identification of soil quality has become a hot research topic.

Remote sensing data has the advantages of low cost, wide coverage, and strong periodicity, making it an indispensable data source for soil quality information identification [12]. Studies have shown that near-infrared spectroscopy, which has a direct spectral response to soil properties such as organic carbon and water content, is a sensitive spectral segment for predicting soil quality [13]. Moreover, the multiple spectral combinations of visible light and near-infrared can more accurately monitor soil's physical, chemical, and biological characteristics [14]. Based on this, many scholars have realized soil quality prediction by using remote sensing images with different resolutions [15,16]. In addition, many scholars extracted remote sensing spectral indicators from crops for growth monitoring and yield estimation, which is also a good reflection of soil quality [17,18]. Among them, the vegetation index was widely used. For example, Dedeoğlu et al. [19] extracted the normalized different vegetation index (*NDVI*), red-edge optimized soil-adjusted vegetation index (*RE-OSAVI*), and red-edge modified chlorophyll absorption in reflectance index (*REMCARI*) from Sentinel images to achieve the classification of soil productivity. Duan et al. [20] integrated multi-source remote sensing data and created a cultivated land quality evaluation system using the *NDVI* index. However, the assessment based on crop remote sensing can only indirectly obtain soil quality status, ignoring the natural attributes and human impact of cultivated land quality. A single data source leads to limited identification accuracy, so the results lack explanatory power.

At present, many scholars try to apply multiple types of data to the soil evaluation field. For example, Yang et al. [21] and Sciortino et al. [22] realized a land productivity assessment by integrating thematic maps and remote sensing image data. Pullanagari et al. [23], Liu et al. [24], and Binte Mostafiz et al. [25] fused with satellite indicators and topographic indices to assess the soil quality of agricultural land. Shi et al. [26] considered human activities and constructed an evaluation framework of cultivated land quality based on resource-asset-capital attributes. In general, the soil quality evaluation based on multi-source data had achieved good results. Still, the current research mostly used a linear combination between single-temporal remote sensing data and other natural data to evaluate the soil quality, which contains limited spectral information and ignores the influence of human factors and the nonlinear characteristics of soil quality. It is rarely reported to realize the automatic identification of soil quality grade by fusing multi-source and multi-temporal data, which needs further research and exploration.

This paper took the Shandong Province as the study area, and aimed to use characteristic variables derived from multi-source data to identify the soil quality information and provide technical support for the utilization and management of cultivated land resources. The specific objectives were to (1) fuse multi-source data based on the Pressure-State-Response (P-S-R) framework to construct characteristic indicators and create a new and automatic soil quality assessment system through the random forest algorithm; (2) analyze the effect of rotation zoning based on crop maturity system on soil quality identification; and (3) fuse remote sensing information through principal component analysis and analyze the effect of multi-temporal remote sensing on soil quality identification.

## 2. Materials and Methods

### 2.1. Study Area

According to the yearbook [27], Shandong Province is located in the eastern part of mainland China, in the North China Plain (34°23′–38°24′N, 114°48′–122°42′E). It is 721.03 km from east to west and 437.28 km from north to south, with a total area of 155,800 square kilometers, and contains 16 cities and 137 counties (Figure 1). On the basis of the Koppen-Geiger climate classification system, it has a cold climate with dry winter and hot summer (DWA) class [28]. The annual average temperature is 11–14 °C. There are sufficient light resources and the annual average light hours are 2290–2890 h. The annual average precipitation is 550–959 mm, decreasing from southeast to northwest. Mountains and hills account for about 29.98% of the province, and plains account for 65.56%. The areas with a slope less than 2° are concentrated in the western plain of Shandong Province, accounting for 71.02%. However, the central and eastern regions generally have higher slopes, and the highest elevation is 1532.7 m. Based on the World Reference Base (WRB) classification [29], fluvisols, leptosols, regosols and luvisols, alisols, and retisols are the main soil types of cultivated land in Shandong Province. Luvisols, alisols, and retisols are widely distributed in the eastern and central-southern hills of Shandong Province, and fluvisols, leptosols, and regosols are mostly distributed in the western plain of Shandong Province, showing a good regional distribution. The plough layer texture is mainly loam and sandy loam, the soil texture profile is mainly loamy and intercalated clay soil, and the plough layer is relatively deep. Shandong is a traditional agricultural province; the cultivated land area is 7.59 million hectares. There is more cultivated land in the western and northern plains, mainly irrigated land, and less cultivated land in the eastern and southern hill areas, mainly dry land. It has a long farming history and high farmland reclamation rate, a complete range of plant industries, and a wide variety of crops. The crop maturity system is one to two crops a year, the double-season crop is mainly wheat and corn, and the common one-season crop is cotton and peanut. According to statistics, the cultivated land irrigation area is 5.191 million hectares in Shandong Province, the power of agricultural machinery is 19.542 kW/ha, and the amount of chemical fertilizer is 847.621 kg NPK/ha. The intensity and level of cultivated land development and utilization rank at the forefront of the country, belongs to intensive agricultural areas with high input and high output, and the agricultural mechanization and modernization are high [30].

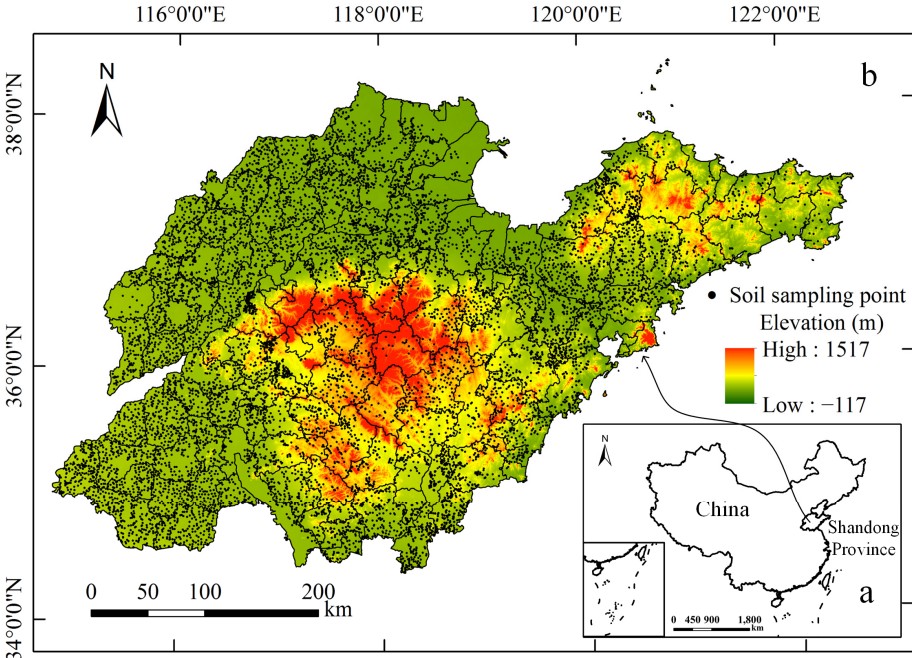

**Figure 1.** Distribution map of study area and soil sampling points. (**a**) China; (**b**) Shandong Province.

### 2.2. Soil Sampling, Analysis, and Data Transformation

(1)    Soil Sampling and Analysis

Related data come from the cultivated land quality grade evaluation project of Shandong Province in 2017. Based on the principle of universality, representativeness, and uniformity, we took the soil species as the basic unit and tried to select fields with similar crop types. Then, soil sampling points were generally arranged according to the area of each point is about 250 m × 250 m. At the same time, combined with geographical location and utilization status, 9673 points were obtained in the end (Figure 1). After the harvest of autumn crops, the field investigation was carried out simultaneously with the county as the basic unit by the same soil sampling methods and test standards. Since farming, fertilization, and other agricultural measures are often carried out in a certain direction. To enhance the representativeness of mixed soil samples, 15–20 small soil sample points were selected uniformly and randomly near the sampling points by the "S" route, and the sampling depth was up to 20 cm. A measure of 1.5 kg soil samples were retained by multi-point mixed sampling and quartile method, and the surrounding geographical location, soil properties, farmland facilities, and other information were recorded simultaneously [31].

The soil samples collected in the field were dried, crushed, and sieved to pass a 2 mm sieve and then analyzed in the laboratory. The test method was shown in Supplementary Table S1 in Supplementary Materials. Then we used the pauta criterion (3σ criterion) to remove outliers, and used the inverse distance weighted (IDW) method to interpolate to be consistent with the resolution of remote sensing images. The IDW takes the distance between interpolation points and sample points as the weight for the weighted average. The distance is closer, the greater the weight given to sample points. This method has the advantages of simple principle, fast calculation speed, intuition, and efficiency, so it is widely used in soil nutrient interpolation [32].

(2)    Basic Maps and Processing

At the same time, we collected and compiled basic maps related to cultivated land quality evaluation in Shandong Province. Such as soil maps, administrative maps, geomorphic maps, irrigation zoning maps, etc. Relevant thematic maps were mainly compiled from the survey results of Counties and Districts in Shandong Province, with a scale of 1:500,000. The basic information of maps is shown in Supplementary Table S1 in Supplementary Materials.

We extracted the relevant information from basic maps and screened it through the Delphi and hierarchical cluster methods. The evaluation factors are shown in Supplementary Table S1. Among them, the hierarchical cluster method is used to screen quantitative indicators. By clustering and merging similar indicators, it helps to select relatively independent leading factors. Delphi method is used to screen qualitative indicators and determine the final indicators according to the experience [33]. Then, the qualitative indicators were quantified by the Delphi method, and interpolated by the IDW method to be consistent with the resolution of remote sensing images.

### 2.3. Acquisition and Processing of Remote Sensing Images and Other Thematic Data

(1)    Acquisition and Processing of Remote Sensing Images

Remote sensing data used a MOD13Q1 dataset of 16-day maximum synthesis from LAADS DAAC (https://ladsweb.modaps.eosdis.nasa.gov, accessed on 1 January 2022), with a spatial resolution is 250 m. A total of 23 images from 2017 to 2018 were used in the study. In addition, the GlobeLand30 global surface cover data in 2020 were selected from China National Geographic Information Center (http://www.globallandcover.com, accessed on 1 January 2022) to assist in the extraction of cultivated land.

As for remote sensing data, the MODIS Reprojection Tool (version 4.1; https://lpdaac.usgs.gov/tools/modis_reprojection_tool, the gdalwarp command from the Geospatial Data Abstraction Library (GDAL) library might be preferred, accessed on 1 January 2022) was used for image mosaic, band screening, and projection conversion. ArcGIS 10.2 software (ESRI, Redlands, CA, USA) was used for image clipping, ENVI 5.1 software (Exelis Visual

Information Solutions, Boulder, CO, USA) was used for band synthesis, HANTS method was used for smoothing processing, and *NDVI* time series data were obtained for the extraction of cultivated land.

(2)    Acquisition and Processing of Thematic Data

Slope data were from the SRTMSLOPE product in Geospatial Data Cloud (http://www.gscloud.cn, accessed on 15 January 2022). Precipitation and temperature data were from China's annal surface climatic data set in China Meteorological Data Network (http://data.cma.cn, accessed on 15 January 2022). Agricultural statistical data were obtained from statistical yearbooks of cities and counties and statistical bulletins of national economic and social development [27,30].

As for other thematic data, graphic processing was carried out first. The slope map was obtained by cutting the slope product. In order to reduce the accidental error, as for meteorological maps, we selected the average meteorological factors values of 121 stations in the study area in recent five years (2015–2019) to interpolate by the IDW method. Additionally, for agricultural statistical maps, we linked each county's average agricultural input indicators in recent five years (2015–2019) to the vector map. Then quantify the thematic map, in which the slope was reclassified and quantified according to relevant technical regulations [31]. The meteorological and agricultural statistical maps were standardized by the z-score method [34], and divided into five levels of "rich, relatively rich, medium, relatively lacking, and lacking" by the equidistant method. The indicators were quantified by the Delphi method, and then rasterized. Finally, the resolution of all raster images was resampled to 250 m by the bilinear interpolation method.

### 2.4. Methods

This study introduced the P-S-R framework to construct soil quality characteristic indicators fusing multi-source data and multi-temporal crop remote sensing information. According to the acquisition of soil state indicators, it was divided into three types: no distinction between crop cover types, distinction between crop cover types, and fusion of multi-temporal data types. Training points of different grades were selected based on the evaluation results, and the identification of cultivated land quality grades was realized by the random forest method. The specific technical route is shown in Figure 2.

#### 2.4.1. Cultivated Land Extraction Based on Remote Sensing

Based on the crop ripening system and spectral curve of ground objects in the study area, the cultivated land was divided into one-season and double-season crop areas, and sample points were arranged with the help of GlobeLand30 data and Google Earth images. Through the random forest algorithm provided by ENMAP-BOX software (version 3.0, EOC of DLR, Cologne, Germany) [35], the cultivated land information was extracted by long time series MODIS-*NDVI* data, and the area extraction accuracy was analyzed by statistical yearbook.

#### 2.4.2. Soil Quality Evaluation Based on Geographic Information System (GIS)

Firstly, the weight ($C_i$) was determined by the analytic hierarchy process (AHP) method. AHP is a decision-making method combining qualitative and quantitative, which can be used to determine the weight of evaluation factors [36]. Secondly, the membership function and membership degree ($F_i$) were determined by the Delphi method and fuzzy statistical method. The fuzzy statistical method is a quantitative method based on the membership degree theory of fuzzy mathematics, which can be used for the normalization of indicators [2]. Thirdly, overlaying each indicator layer, the soil quality index (*SQI*) was calculated by the Formula (1), and soil types were divided into high (grade 1 and 2), medium (grade 3 and 4), and low (grade 5 and 6) by the natural breakpoint method [31,37].

$$SQI = \sum_{i=1}^{n}(C_i \times F_i); \tag{1}$$

where $C_i$ represents the combined weight of the *i*th indicator, and $F_i$ represents the membership degree of the *i*th indicator.

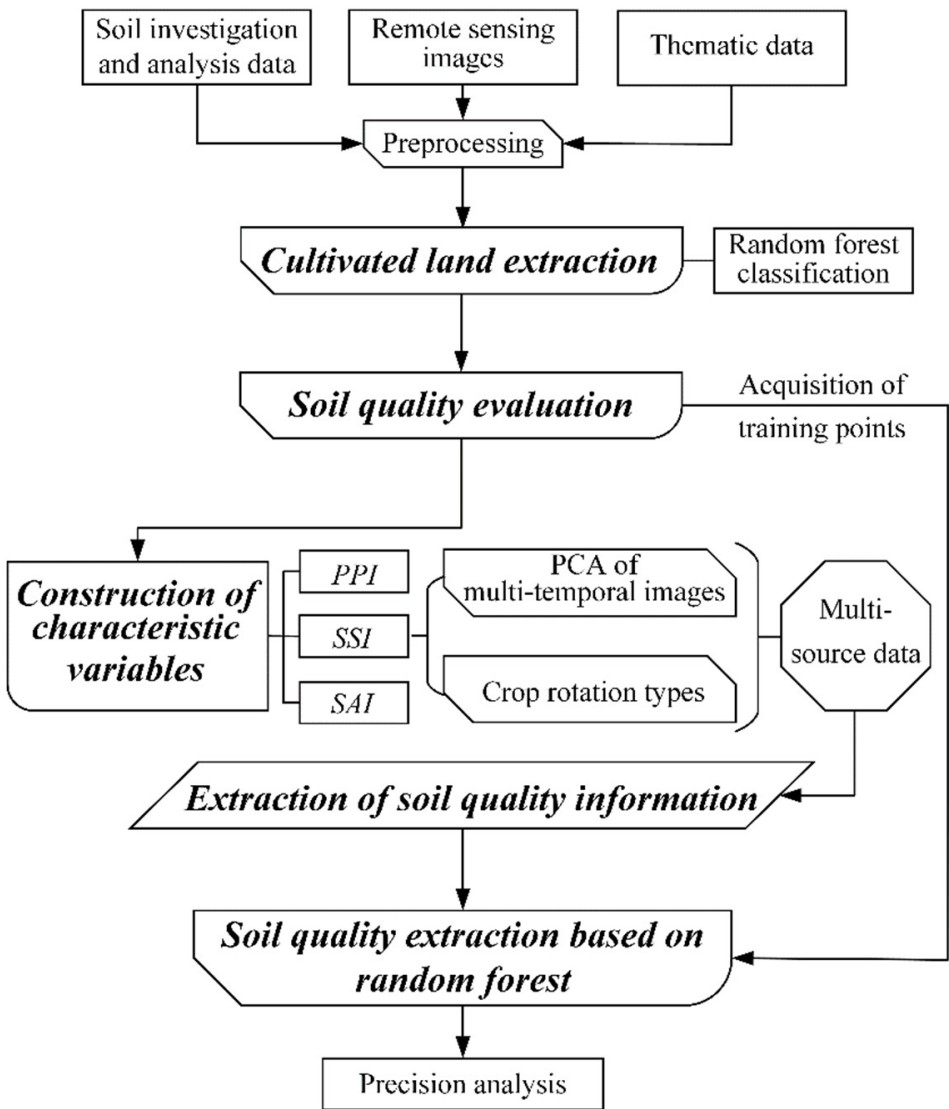

**Figure 2.** Technology roadmap. Note: PCA: principal component analysis, *PPI*: production press indicators, *SSI*: soil status indicators, *SAI*: social action indicators.

### 2.4.3. Construction of Soil Quality Characteristic Indicators Based on P-S-R Framework

The Pressure-State-Response model, namely the P-S-R model, is suitable for analyzing the relationship between environmental pressure, current situation, and response and land assessment [38]. Based on the P-S-R framework theory, this study selected characteristic indicators to construct the cultivated land quality evaluation system from the environmental pressure, the existing quality status, and the behavior response of cultivated land users.

(1) Production Press Indicators

Production press indicators (*PPI*) were designed to reflect the natural endowment and limiting factors of soil in the study area, mainly obtained from thematic maps. The geomorphic types are complex, and the terrain fluctuates greatly in the study area. Slope is an important factor limiting the development of soil quality. In addition, there are significant differences in temperature and precipitation in the study area due to its vast territory, which leads to spatial variation of soil quality to a certain extent. Therefore, slope, annual mean precipitation, and annual mean temperature were selected as *PPI*.

(2)    Soil Status Indicators

Soil status indicators (*SSI*) were designed to reflect the soil fertility and degradation risk in the study area, mainly obtained from remote sensing data. Normalized difference vegetation index (*NDVI*) can reflect vegetation coverage and biomass, indirectly reflect soil fertility, and can be used as a soil fertility indicator. Difference vegetation index (*DVI*) is sensitive to water change and can reflect soil moisture, which can be used as a soil moisture indicator. Ratio vegetation index (*RVI*) is sensitive to soil ecological environment stress and can reflect the degree of soil degradation, which can be used as a soil degradation indicator [39]. The specific description of indicators is shown in Table 1.

**Table 1.** Description of soil status indicators.

|  | Indicators | Expression | Reference |
|---|---|---|---|
| Soil fertility indicator | *NDVI* | $\frac{\rho_{NIR}-\rho_R}{\rho_{NIR}+\rho_R}$ |  |
| Soil moisture indicator | *DVI* | $\rho_{NIR}-\rho_R$ | [39] |
| Soil degradation indicator | *RVI* | $\frac{\rho_{NIR}}{\rho_R}$ |  |

Note: $\rho_R$ and $\rho_{NIR}$ are the red and near-infrared bands of MODIS data, respectively.

(3)    Social Action Indicators

Social action indicators (*SAI*) were designed to reflect the agricultural input and management level of soil in the study area, mainly obtained from statistical data. The input of basic agricultural factors such as labor, agricultural machinery, irrigation equipment, and fertilizer indirectly reflects decision-makers' behavior response to cultivated land and is an important human factor affecting soil quality.

This study constructed *SAI* based on statistical data of counties in recent five years (2015–2019), including agricultural labor indicator (*ALI*), agricultural mechanization indicator (*AMI*), agricultural irrigation indicator (*AII*), and agricultural fertilizer indicator (*AFI*). They represent the average level of the rural labor resources, the total power of agricultural machinery, the effective irrigated area, and the chemical fertilizer consumption. The specific description of indicators is shown in Table 2.

**Table 2.** Description of social action indicators.

| Indicators | Expression | Units | Reference |
|---|---|---|---|
| *ALI* | $\frac{Rural\ labor\ resources}{Total\ population}$ | % |  |
| *AMI* | $\frac{Total\ power\ of\ agricultural\ machinery}{Cultivated\ land\ area}$ | kW/ha | [30] |
| *AII* | $\frac{Effective\ irrigated\ area}{Cultivated\ land\ area}$ | % |  |
| *AFI* | $\frac{Consumption\ of\ chemical\ fertilizer}{Cultivated\ land\ area}$ | kg NPK/ha |  |

Note: Total population refers to the total number of people alive at a certain point of time within a given area. Cultivated land area refers to the area extracted by remote sensing. Rural labor resources refer to those who can participate in production and business activities above working age in the rural population. Total power of agricultural machinery refers to total mechanical power of machinery used in farming, forestry, animal husbandry, and fishery. Effective irrigated area refers to cultivated area with some water sources, which is smooth, provided with irrigation engineering or equipment and capable of normal irrigation in the latter half of general year. Consumption of chemical fertilizer refers to the quantity of chemical fertilizers applied in agriculture in a year, including nitrogenous fertilizer, phosphate fertilizer, potash fertilizer, and compound fertilizer, which is calculated to convert the gross weight into weight containing 100% effective component.

2.4.4. Extraction of Soil Status Indicators under Three Situations

To distinguish the influence of crop remote sensing on soil quality grade identification in different situations, this study divided *SSI* into three types: no distinction between crop cover types, distinction between crop cover types, and fusion of multi-temporal data types.

(1)    No distinction between crop cover types (Method A)

It is composed of *SSI* in a single period. According to the time series curves of *NDVI*, Synthetic period with the most abundant biomass was selected as the sensitive period to highlight crop information and reflect the soil quality state more fully. The *SSI* was constructed based on a single sensitive period of MODIS synthetic images.

(2)    Distinction between crop cover types (Method B)

It is composed of *SSI* in two periods. The crop cover area was divided into one season and double season crop areas according to the extraction results of cultivated land, and *SSI* was constructed based on two sensitive period of MODIS synthetic images, respectively.

(3)    Fusion of multi-temporal data types (Method C)

It is composed of *SSI* in multiple periods. Principal component analysis (PCA) is a popular data dimension reduction method. It is a multivariate statistical method that converts multiple variables into a few principal components under the premise of losing as little information as possible [40]. This study took the county as the unit to calculate the weighted average value of *SSI* and used the *SSI* at the sensitive period of MODIS synthetic images as the baseline to conduct correlation analysis. The remote sensing images with significant correlation and correlation coefficient greater than 0.3 were screened. After that, we conducted KMO and Bartlett tests on the selected factors after correlation analysis. When KMO is greater than 0.5 and the significant level of the Bartlett test is less than 0.05, it is considered to pass the test [41]. Finally, the *SSI* fused with multi-temporal information (*MT-SSI*) was constructed by PCA.

2.4.5. Identification of Soil Quality Grade Based on Multi-source Data

(1)    Acquisition of Training Samples

The study area was divided into 10 km $\times$ 10 km grids, and the soil quality evaluation grade based on GIS was extracted by the grid center point to obtain training points of different grades.

(2)    Identification of Soil Quality Grade

Based on training points and multi-source data in the above three situations, the random forest classification method provided by ENMAP-BOX software was used to identify the soil quality grade. The tree node (ntree) was set to 500, and the feature number (mtry) was set to "square root". The median filtering method (the convolution kernel size was set to 5 $\times$ 5) was used for smoothing processing to enhance the classification effect and eliminate salt and pepper noise.

(3)    Verification of Identification Accuracy

By comparing the identification results based on multi-source data and the evaluation results based on GIS, the spatial distribution accuracy was verified by comparing the spatial distribution differences of soil types in different grades, and the area identification accuracy was verified by comparing the area differences of soil types in different grades. The verification point identification accuracy took the sample points of cultivated land extraction as the verification points, the evaluation grade as the real data and the identification grade as the classification data. Then the confusion matrix was constructed to calculate the user accuracy (UA), producer accuracy (PA), overall Accuracy (OA), and Kappa coefficient to analyze the identification accuracy of high-, medium-, and low-grade soil types.

### 3. Results and Analysis

*3.1. The Results of Cultivated Land Information Extraction Based on Remote Sensing*

Figure 3a shows the distribution of sample points and cultivated land extraction results, the sample points include 179 one-season crop points and 275 double-season crop points. The *NDVI* spectral curves of different ground objects are different (Figure 3b), in which the double-season crop has typical bimodal characteristics, reaching the peak values

in mid-April (point A) and mid-August (point B), respectively; the one-season crop is a unimodal curve, peaked in mid-August (point C), and has a good distinction with other ground objects. It has been verified that the extracted accuracy of cultivated land area is 87.75%. The one-season crop area is mainly distributed in the eastern and central-southern hills of Shandong Province, and the double-season crop area is mainly concentrated in the western plain of Shandong Province and Jiaolai Plain, which are widely distributed and can better meet the research needs.

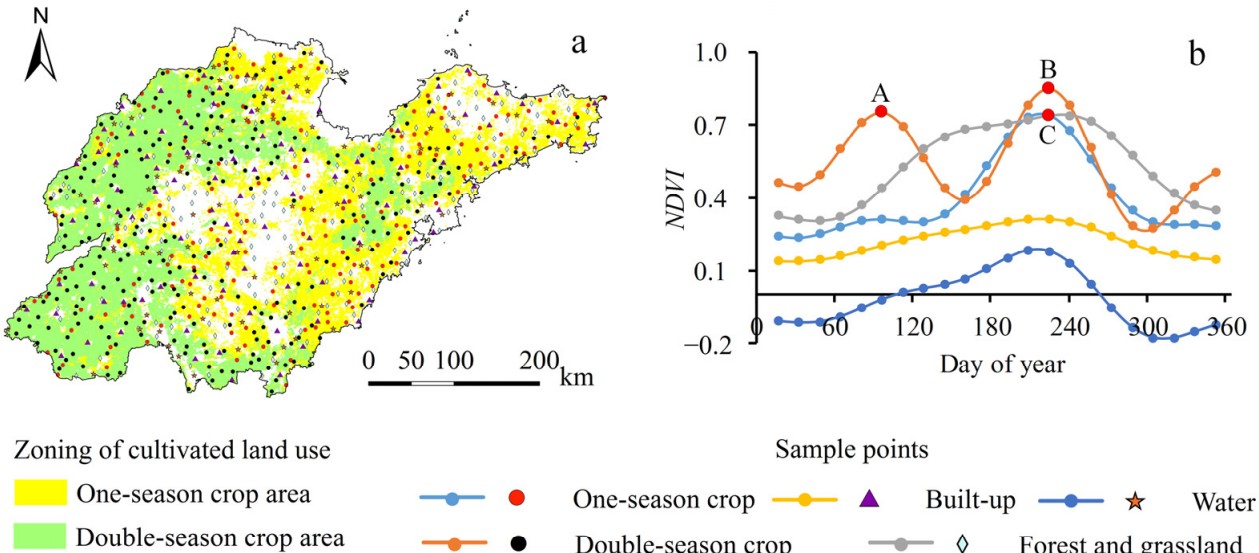

**Figure 3.** Extraction results of cultivated land information. (**a**) Spatial distribution of sample points and cultivated land; (**b**) *NDVI* spectral curves of different ground objects. Note: A and B are the *NDVI* peaks of double-season crop, respectively, and C is the *NDVI* peak of one-season crop.

### 3.2. The Results of Soil Quality Evaluation Based on GIS

The descriptive statistical characteristics of soil quality evaluation indicators and their weight values are shown in Supplementary Table S2 in Supplementary Materials. According to the typical characteristics and agricultural background of the cultivated land system in Shandong Province, six physical indicators, five chemical indicators, and two biological and environmental indicators were selected. The evaluation results of cultivated land quality are shown in Figure 4. The spatial distribution of different soil quality grades is related mainly to topographic factors. The high grades are mostly distributed in the northwestern plain of Shandong Province, Jiaolai Plain, and the southern part of the central and southern hills of Shandong Province, the medium grades are mostly distributed in the western plain of Shandong Province and Yellow River Delta, and the low grades are mostly distributed in the central and southern hills of Shandong Province and Jiaodong hills.

### 3.3. Construction and Analysis of Soil Quality Characteristic Indicators
3.3.1. Construction Results of Soil Quality Indicators under Three Multi-Source Data Situations

According to the *NDVI* spectral curves of one-season crops and double-season crops (Figure 3b), point A (97–112 days, from April 7 to April 22) and point B (225–240 days, from August 13 to August 28) are the two sensitive periods corresponding to double-season crops, respectively, and point C (225–240 days, from August 13 to August 28) is the sensitive period corresponding to one-season crops. Accordingly, the soil quality characteristic indicators constructed under three different situations are shown in Table 3.

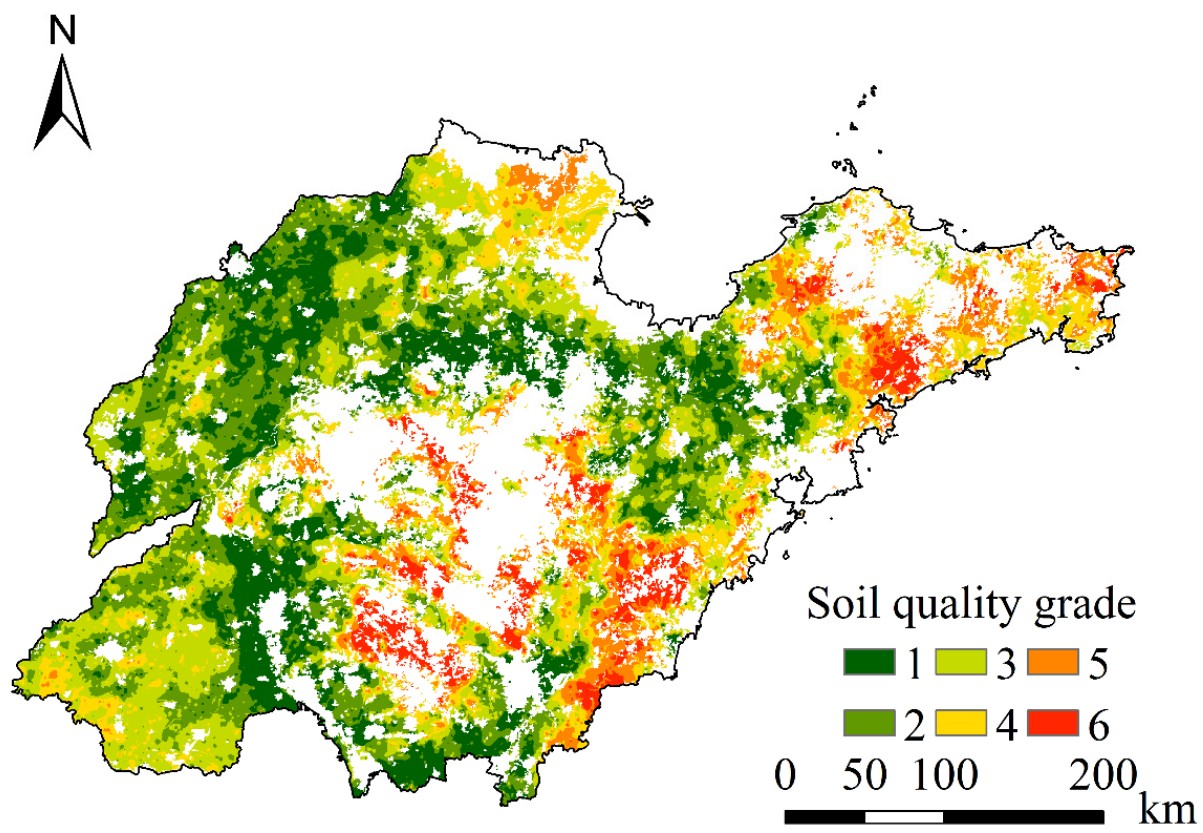

**Figure 4.** Evaluation results of cultivated land quality.

**Table 3.** Characteristic indicators under three multi-source data situations.

|  |  | Method A | Method B | | Method C | |
|---|---|---|---|---|---|---|
|  |  |  | One-Season Crop Area | Double-Season Crop Area | One-Season Crop Area | Double-Season Crop Area |
| *PPI* |  | Slope, Annual average precipitation, Annual average temperature | | | | |
| *SSI* | Soil fertility indicator | $NDVI_{225-240}$ | $NDVI_{225-240}$ | $NDVI_{097-112}$ | $MT\text{-}NDVI_{225-240}$ | $MT\text{-}NDVI_{097-112}$ |
|  | Soil moisture indicator | $DVI_{225-240}$ | $DVI_{225-240}$ | $DVI_{097-112}$ | $MT\text{-}DVI_{225-240}$ | $MT\text{-}DVI_{097-112}$ |
|  | Soil degradation indicator | $RVI_{225-240}$ | $RVI_{225-240}$ | $RVI_{097-112}$ | $MT\text{-}RVI_{225-240}$ | $MT\text{-}RVI_{097-112}$ |
| *SAI* |  | *ALI, AMI, AII, AFI* | | | | |

Note: Method A, Method B, and Method C are the no distinction between crop cover types, distinction between crop cover types, and fusion of multi-temporal data types, respectively. *PPI*, *SSI*, and *SAI* are the production press indicators, soil status indicators, and social action indicators, respectively. $NDVI_{097-112}$, $DVI_{097-112}$, and $RVI_{097-112}$ are the *SSI* constructed by the synthetic images on 97–112 days; $NDVI_{225-240}$, $DVI_{225-240}$, and $RVI_{225-240}$ are the *SSI* constructed by the synthetic images on 225–240 days; $MT\text{-}NDVI_{097-112}$, $MT\text{-}DVI_{097-112}$, and $MT\text{-}RVI_{097-112}$ are the fusion of multi-temporal *SSI* constructed based on synthetic images of 97–112 days; $MT\text{-}NDVI_{225-240}$, $MT\text{-}DVI_{225-240}$, and $MT\text{-}RVI_{225-240}$ are the fusion of multi-temporal *SSI* constructed based on synthetic images of 225–240 days. *ALI*, *AMI*, *AII*, and *AFI* are the agricultural labor indicator, agricultural mechanization indicator, agricultural irrigation indicator, and agricultural fertilizer indicator, respectively.

As for *SSI*, the synthetic remote sensing data of 225–240 days were used for Method A. The synthetic remote sensing data of 225–240 days and 97–112 days were used for one-season and double-season crop areas in Method B. Based on synthetic remote sensing data of 225–240 days and 97–112 days, respectively, other images with strong correlation were selected to form a multi-temporal data set by PCA in Method C.

### 3.3.2. The Analysis of PPI and SAI

The descriptive statistical characteristics of *PPI* and *SAI* are shown in Supplementary Table S3 in Supplementary Materials. At the same time, Figure 5 shows the spatial distribution after *PPI* and *SAI* pretreatment. From the perspective of *PPI*, the natural resource endowment is good. The mean value of slope is 1.36°, the mean value of annual average precipitation is 722.43 mm, and the mean value of annual average temperature is 13.98 °C. Cultivated land is generally gentle with little slope fluctuation, which is suitable for agricultural farming. Moderate precipitation and temperature ensure water and heat resources needed for crop growth [42]. From the perspective of *SAI*, the mean value of *ALI* and *AII* are 47.59% and 62.64%, respectively, the mean value of *AMI* is 14.34 kW/ha, and the mean value of *AFI* is 653.14 kg NPK/ha. The agricultural resources input and management level of cultivated land is higher, which is conducive to the sustainable development of agriculture and the increase in crop production and income [43]. The overall *SAI* shows the spatial distribution law of high in the west and low in the east. Among them, the input of agricultural labor resources is mainly concentrated in the northwest and south, the input of agricultural fertilizer is mainly concentrated in the northeast and south, and the level of agricultural mechanization and irrigation is significantly higher in the plain area.

### 3.3.3. The Construction Results of MT-SSI

Table 4 shows the construction results of *MT-SSI*. After PCA, *MT-SSI* in different crop rotation areas contains two to three principal components, and each principal component contains five to eight temporal remote sensing information. The cumulative variance contribution rates are greater than 70.25%, which contains most of information in multi-temporal data. After KMO and Bartlett test, KMO > 0.607, Sig. = 0.000, the PCA results are ideal. The specific principal component analysis expression is shown in Supplementary Table S4 in Supplementary Materials. Finally, through grid operation, the descriptive statistical characteristics of SSI are shown in Supplementary Table S3 in Supplementary Materials. The mean soil fertility indicators are between 0.68 and 0.79, the mean soil moisture indicators are between 0.29 and 0.36, and the mean soil degradation indicators are between 2.99 and 8.54. At the same time, the spatial distribution of SSI is shown in Figure 6.

### 3.4. Identification Results of Soil Quality Grade

#### 3.4.1. The Analysis of Spatial Distribution Accuracy for Soil Quality Grade Identification

Figure 7a is the spatial distribution map of training points, which shows complete grades and uniform distribution, it can meet the research needs. By comparing the evaluation results based on GIS (Figure 4) and the identification results based on multi-source data (Figure 7b–d), it was found that the soil quality grade has similar spatial distribution rules, that is, the medium and high grades were mostly concentrated in plain areas, while the low grades were mostly concentrated in hilly areas. From the spatial distribution of each grade, the identification results in Method C have the highest similarity with the evaluation results, followed by Method B, and the difference in Method A is significant.

#### 3.4.2. The Analysis of Area Accuracy for Soil Quality Grade Identification

Table 5 compares the area of each grade in the evaluation results based on conventional GIS and the identification results based on multi-source data. Among them, the error of Method A is the largest, which the maximum area ratio errors of grades 6 and 3 are 12.45% and 4.84%, respectively. After distinguishing crop rotation types, the area ratio errors of grades 6 and 3 are less than 4.83% and 2.68%, respectively, indicating that the area identification accuracy is significantly improved. The area identification accuracy of Method C is the highest, and the area ratio errors of grades 6 and 3 are less than 1.31% and 1.13%, respectively.

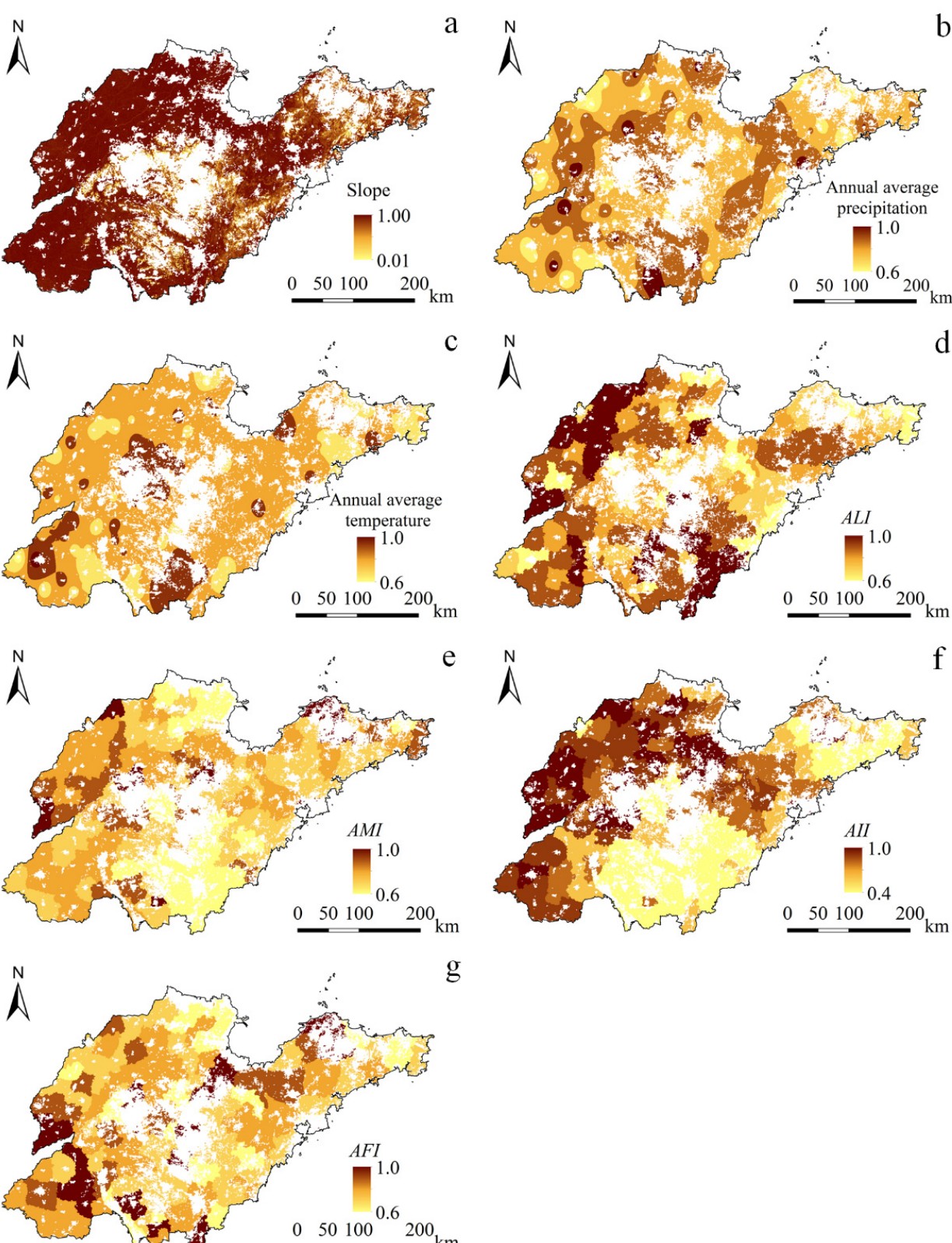

**Figure 5.** Spatial distribution map of production press indicators and social action indicators after membership processing. (**a**–**c**) are production press indicators; (**d**–**g**) are social action indicators.

**Table 4.** Principal component analysis results of soil state indicators.

| Scheme | Indicator | Principal Component | Synthetic Period of Images | Cumulative Variance Contribution Rate | KMO | Sig. |
|---|---|---|---|---|---|---|
| One-season crop area | *MT-NDVI* | PC1 | 177–192, 193–208, 209–224, 225–240, 241–256 | 55.33% | 0.610 | 0.000 |
| | | PC2 | | 83.49% | | |
| | *MT-DVI* | PC1 | 177–192, 193–208, 209–224, 225–240, 241–256, 257–272 | 53.08% | 0.681 | 0.000 |
| | | PC2 | | 82.93% | | |
| | *MT-RVI* | PC1 | 193–208, 209–224, 225–240, 241–256, 257–272 | 55.75% | 0.607 | 0.000 |
| | | PC2 | | 81.18% | | |
| Double-season crop area | *MT-NDVI* | PC1 | 065–080, 081–096, 097–112, 113–128, 129–144, 273–288, 289–304, 337–352 | 52.42% | 0.724 | 0.000 |
| | | PC2 | | 70.25% | | |
| | | PC3 | | 83.57% | | |
| | *MT-DVI* | PC1 | 065–080, 081–096, 097–112, 113–128, 129–144, 145–160, 177–192 | 57.11% | 0.736 | 0.000 |
| | | PC2 | | 71.43% | | |
| | *MT-RVI* | PC1 | 065–080, 081–096, 097–112, 113–128, 129–144, 273–288, 289–304, 337–352 | 52.33% | 0.714 | 0.000 |
| | | PC2 | | 69.69% | | |
| | | PC3 | | 83.50% | | |

Note: *MT-NDVI*, *MT-DVI*, and *MT-RVI* are the fusion of multi-temporal soil fertility indicator, soil moisture indicator, and soil degradation indicator, respectively. $PC_1$, $PC_2$, and $PC_3$ represent principal components 1, 2, and 3, respectively.

**Table 5.** Area comparison of soil quality evaluation and identification.

| Grade | | Evaluation | | Method A | | | Method B | | | Method C | | |
|---|---|---|---|---|---|---|---|---|---|---|---|---|
| | | Area Ratio % | | Area Ratio % | | Difference | Area Ratio % | | Difference | Area Ratio % | | Difference |
| High | 1 | 47.31 | 19.13 | 52.15 | 11.52 | 7.61 | 48.17 | 15.16 | 3.97 | 47.78 | 18.64 | 0.49 |
| | 2 | | 28.18 | | 40.63 | 4.84 | | 33.01 | 0.86 | | 29.14 | 0.47 |
| | | | | | | 12.45 | | | 4.83 | | | 0.96 |
| Medium | 3 | 37.98 | 23.43 | 37.11 | 21.33 | 2.10 | 39.80 | 22.03 | 1.40 | 36.85 | 22.60 | 0.83 |
| | 4 | | 14.55 | | 15.78 | 0.87 | | 17.77 | 1.82 | | 14.25 | 1.13 |
| | | | | | | 1.23 | | | 3.22 | | | 0.30 |
| Low | 5 | 14.71 | 9.88 | 10.74 | 8.05 | 1.83 | 12.03 | 8.04 | 1.84 | 15.37 | 11.19 | 1.31 |
| | 6 | | 4.83 | | 2.69 | 3.97 | | 3.99 | 2.68 | | 4.18 | 0.66 |
| | | | | | | 2.14 | | | 0.84 | | | 0.65 |
| Summation | | 100.00 | | 100.00 | | - | 100.00 | | - | 100.00 | | - |

Note: Method A, Method B, and Method C are the no distinction between crop cover types, distinction between crop cover types, and fusion of multi-temporal data types, respectively.

### 3.4.3. The Analysis of Verification Points Accuracy for Soil Quality Grade Identification

Table 6 shows the identification accuracy of soil quality grade based on verification points. After distinguishing crop rotation types and fusing multi-temporal information, from the perspective of the UA and PA, the mean UA of high, medium, and low grades soil types increased from 75.88% to 91.88%, and the mean PA increased from 80.11% to 93.29%; from the perspective of the OA, it increased from 79.18% to 86.12% and 93.65%; from the perspective of the Kappa coefficient, it increased from 0.66 to 0.77 and 0.90.

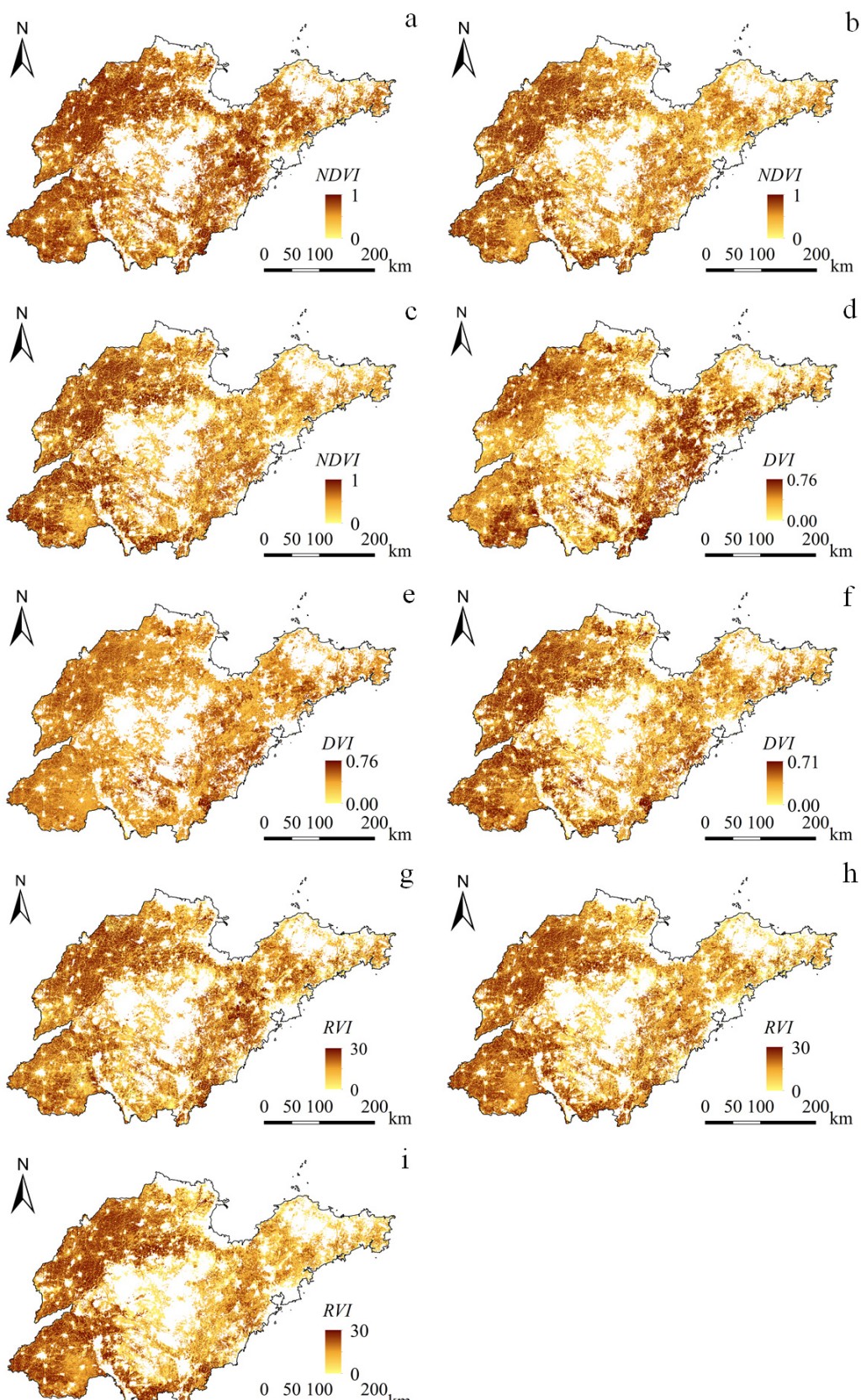

**Figure 6.** The soil status indicators map under three multi-source data situations. (**a**,**d**,**g**) are indicators of the no distinction between crop cover types; (**b**,**e**,**h**) are indicators of the distinction between crop cover types; (**c**,**f**,**i**) are indicators of the fusion of multi-temporal data types.

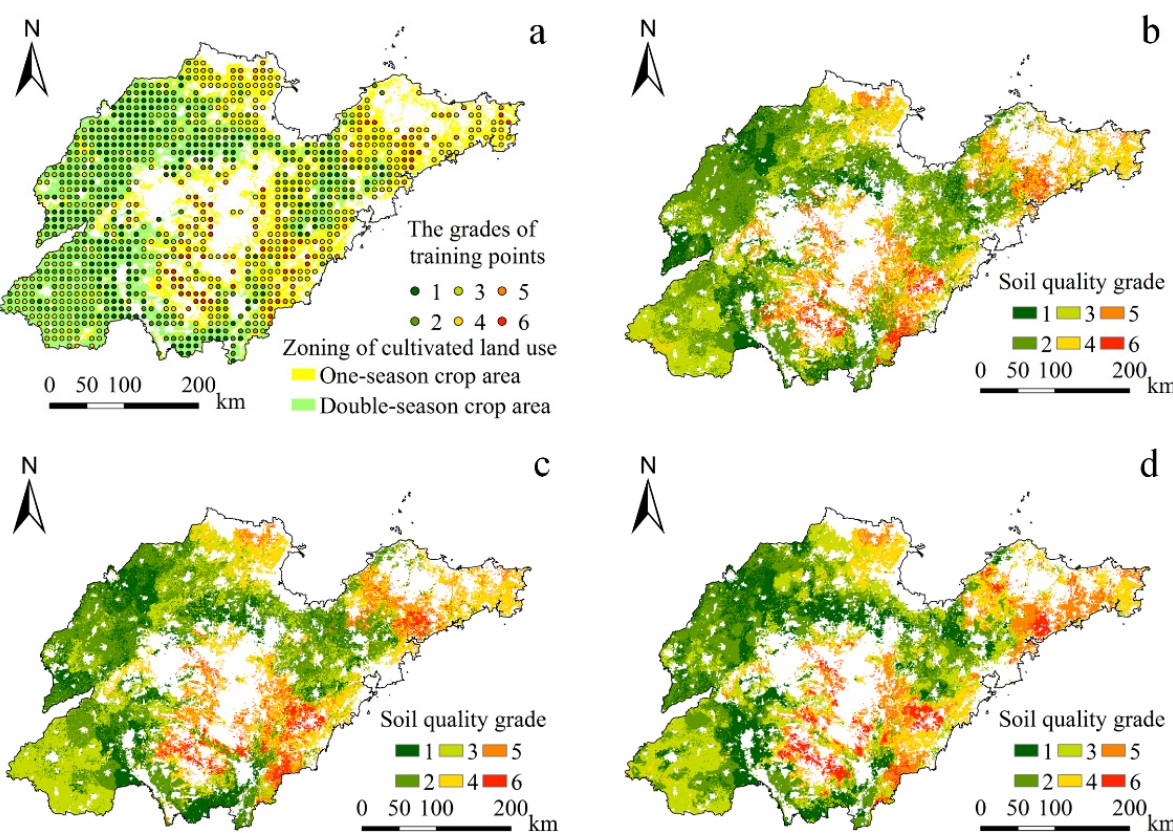

**Figure 7.** Soil quality identification results based on multi-source data. (**a**) Distribution map of training points; (**b**) no distinction between crop cover types; (**c**) distinction between crop cover types; and (**d**) fusion of multi-temporal data types.

**Table 6.** Identification accuracy evaluation of verification points.

|  | UA$_{mean}$ | PA$_{mean}$ | OA | Kappa |
|---|---|---|---|---|
| Method A | 75.88% | 80.11% | 79.18% | 0.66 |
| Method B | 84.14% | 86.90% | 86.12% | 0.77 |
| Method C | 91.88% | 93.29% | 93.65% | 0.90 |

Note: Method A, Method B, and Method C are the no distinction between crop cover types, distinction between crop cover types, and fusion of multi-temporal data types, respectively. UA$_{mean}$ and PA$_{mean}$ represent the mean values of user accuracy and producer accuracy, OA represents the overall accuracy, and Kappa represents the value of the Kappa coefficient.

In conclusion, Method C has the highest identification accuracy, followed by Method B, and the effect of Method A is the worst.

## 4. Discussion

(1) By analyzing the multi-temporal *NDVI* spectral curves of different crop rotation systems, we obtained the most abundant period of crop biomass, that is, the mid-April and mid-August of two-season crops and the mid-August of one-season crops. Referring to previous research experience [19,44], we used three vegetation spectral indicators (*NDVI*, *DVI*, and *RVI*) as soil status indicators to construct the soil quality indicator system, which indirectly realized the identification of cultivated land quality. The research shows that crop remote sensing images of several sensitive periods have obvious advantages in soil quality identification, which reflects the application potential of crop remote sensing in cultivated land quality and has important significance for soil quality prediction at a regional scale. However, the lack of relationship

analysis between soil quality and multi-temporal crop spectrum is the deficiency of this study, which needs to be further optimized.

(2) The soil quality grade identification method proposed in this study fuses terrain, meteorological data, remote sensing data, statistical yearbook, etc. Compared with previous studies [21,24,25], it is found that MODIS data synthesized by 16-day maximum not only reduced the influence of clouds, but also ensured the temporal and spatial continuity of earth observation data. It could more completely express the spectral characteristics of crops in each growth cycle and more accurately reflect the cultivated land quality [45,46]. At the same time, incorporating human activity factors into the soil quality identification system through agricultural statistics data can indirectly reflect the agricultural input and management level and help to improve the identification accuracy. The fusion of multi-source and multi-temporal data will be an effective means to identify the cultivated land quality grade.

(3) By comparing the identification results under two situations, after distinguishing crop rotation types, the maximum area ratio error decreased from 4.84% to 2.68%, the overall accuracy of verification points increased by 6.94%, and the identification accuracy of soil quality grade was improved. It is considered that the partition of crop rotation types reduces the spectral confusion problem and enhances the purity of spectral information, thereby improving the identification accuracy [47,48]. However, the identification accuracy of soil quality based on crop rotation zoning is limited. It is necessary to further use higher resolution Sentinel or Landsat data to distinguish crop types. Accurate classification and partition identification based on crops will also be an effective way to improve the identification accuracy of soil quality.

(4) By comparing the identification results between the fusion of multi-temporal data types and the other two situations, it is found that the former results are more similar to the evaluation results. Compared with only distinguishing crop rotation types, the maximum area ratio error decreased from 2.68% to 1.13%, the overall accuracy of verification points increased by 7.53%, and the identification accuracy of soil quality information was significantly improved. It is mainly due to the fusion of multi-temporal remote sensing data through principal component analysis, which makes the spectral information of soil quality more abundant, and changes from static crop state information to dynamic crop spectral characteristics. It effectively avoids the disadvantages of single-temporal image information and being susceptible to external factors, and enhances the stability of remote sensing data sources, thereby improving the identification accuracy of soil quality grade [49,50].

(5) This study used the random forest algorithm to identify soil quality grades. Compared with the evaluation method based on GIS, this method does not require field sampling and indoor laboratory analysis and avoids the large consumption of human, material, and financial resources [51]. It overcomes the dependence of traditional evaluation on different spatial interpolation methods and can more objectively reflect the spatial distribution information of cultivated land quality [52]. Compared with previous studies [21,25,26], the algorithm fully excavates the nonlinear relationship between multi-source data and has the advantages of high generalization performance, strong fault tolerance and anti-interference ability, and good robustness [53]. It avoids the disadvantages of traditional linear combination methods with strong subjectivity, can fully reflect the comprehensive, random, and nonlinear characteristics of soil quality, and is proved to be an effective method for automatic identification of soil quality grade. In the future, the proposed method can be applied to the rapid interpretation of soil quality grade, assist in establishing long-term monitoring, evaluation, and early warning mechanism of cultivated land quality, and guide agricultural management according to local conditions. To avoid the degradation of cultivated land, maintain and improve the cultivated land quality, and ensure the sustainable use of cultivated land resources.

## 5. Conclusions

In this paper, the P-S-R framework was used to construct soil quality characteristic indicators fused with multi-source and multi-temporal data, and the random forest algorithm was used to realize the rapid and accurate identification of soil quality grade information. The main conclusions are as follows:

(1)    The *NDVI* time series curve of double-season crop shows a typical bimodal characteristic, with the peak in mid-April and mid-August, respectively. In comparison, the one-season crop is a unimodal curve, with the peak value in mid-August. Then, through evaluation, the cultivated land quality was divided into three categories (high, medium, and low), with six grades.

(2)    Three different situations were constructed to extract *SSI*. Synthetic images of 225–240 days were used to the no distinction between crop cover types, synthetic images of 225–240 days and 97–112 days were used to the distinction between crop cover types. Additionally, the fusion of multi-temporal data types was based on the two sensitive periods, and other highly correlated images were selected to form a multi-temporal data set. Through principal component analysis, it contains two to three principal components, and each principal component contains five to eight temporal remote sensing information.

(3)    Distinguishing crop rotation types has a significant gain on the identification accuracy of soil quality grade. Specifically, the spatial distribution of soil types is more similar to the evaluation results, the maximum area ratio error decreased from 4.84% to 2.68%, the overall accuracy increased from 79.18% to 86.12%, and the Kappa coefficient increased from 0.66 to 0.77.

(4)    Fusion of multi-temporal remote sensing data is the best method for soil quality information extraction. Specifically, the spatial distribution of soil types is more similar to the evaluation results, the maximum area ratio error decreased from 2.68% to 1.13%, the overall accuracy of verification points increased from 86.12% to 93.65%, and the Kappa coefficient increased from 0.77 to 0.90.

This paper fused multi-source and multi-temporal crop remote sensing data and provided a fast and efficient method for soil quality grade identification based on a random forest algorithm, which provided a new technical means for the utilization and management of cultivated land resources and the sustainable development of agriculture.

**Supplementary Materials:** The following supporting information can be downloaded at: https://www.mdpi.com/article/10.3390/rs14092109/s1, Supplementary Table S1: Description of soil quality indicators; Supplementary Table S2: Descriptive statistical and analysis table of soil quality evaluation indicators; Supplementary Table S3: Descriptive statistical table of soil quality characteristic indicators; Supplementary Table S4: Principal component analysis results of soil state indicators.

**Author Contributions:** Conceptualization, formal analysis, methodology, visualization, and writing-original draft, Y.L.; data curation, software, validation, and writing—review & editing, C.C.; data curation, software, visualization, and validation, Z.W.; investigation, project administration, and resources, T.L. and J.L.; funding acquisition, resources, supervision, and writing—review & editing, G.Z. All authors have read and agreed to the published version of the manuscript.

**Funding:** This research was funded by the National Natural Science Foundation of China (41877003), the Major Scientific and Technological Innovation Project in Shandong Province (2019JZZY010724), and the Funds of Shandong "Double Tops" Program (SYL2017XTTD02).

**Data Availability Statement:** Not applicable.

**Acknowledgments:** Thanks to the NASA MODIS teams for providing MODIS data. Furthermore, we appreciate the editors and reviewers for their constructive comments and suggestions.

**Conflicts of Interest:** The authors declare that they have no conflict of interest.

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
