# Peer review of "Identification of Cultivated Land Quality Grade Using Fused Multi-Source Data and Multi-Temporal Crop Remote Sensing Information"

_remotesensing, doi:10.3390/rs14092109_

Round 1
Reviewer 1 Report
The authors have presented a detailed procedure to identify the soil quality grade using the remote sensing datasets and the Random Forest method. The concept is well formulated but there are some issues that need to address. The specific comments are given below. Accordingly, a major revision of the manuscript has been recommended.
Comments:
- Abstract: Abbreviate P-S-R in L17. In this section after L-17, most of the sentences are long and need Rewriting.
- Introduction (L62-72): Literature related to how remote sensing indices are useful for soil quality study is missing. Describe with examples from NDVI, SAVI, etc..
- What spectral wavelengths are relevant for soil quality assessment. One paragraph on this will be useful to readers.
- At the end of the Introduction, mention the objective of the study.
- Section 2.1 (Fig .1). In L-127: How many sampling points? It is not clear at this stage, what sampling design was adopted and what grid size
- Methods: L-181. Here specify clearly which multi-temporal data types
- Fig 2 and 5: The resolution of these figures need to improve
- L217:221: Which bands were used to derive NDVI, DVI and RVI
- Table 1: How the weights are assigned.
- L304: Fig 4 and Fig 7. What differences exist between them? Is it a repetition of Figure 4.
- L489-491: Rewrite
- The title needs to change as: ...using Fused Multi-source
- Many abbreviations are used in entire ms but never introduced (e.g. P-S-R, NDVI, RVI, MT-NDVI so on…)
- Discussion (L422): long-term series is not the right word here as the authors have used only the 2017-2018 dataset.
Author Response
Thank you for the time and effort spent in reviewing the manuscript. We acknowledge your comments and suggestions very much, which are valuable in improving the quality of our manuscript. Please see the attachment for my responses to your questions and suggestions.

Reviewer 2 Report
Soil quality evaluation is a very important for the management of crop land and it is very difficult work. This paper fused with multi-source and multi-temporal crop remote sensing data and provided a fast and efficient method for soil quality grade identification based on a random forest algorithm. It provided a new technical method for the utilization and management of cultivated land resources and the sustainable development of agriculture.
I just have one comment. It is difficult to distinguish the one-season crop area and double-season crop area on Figures 3 and 7, Please try to use different colors for the two area.
Author Response

(The authors gave the same response as above.)

Reviewer 3 Report
GENERAL
The paper treats about fast, remote-sensing method of discrimination and identification of cultivated land areas of different quality and it proves that the integration of multi-source, and, particularly, remote multi-temporal data is most efficient. It is also important that authors validated their findings on rather extensive and representative set of in-situ soil data.
However, I see many issues regarding the description of study area and methods used in the paper. I had to browse many terms in internet. If we want the paper to be accessible for broader group of readers, and not only those specialised in Remote Sensing, these terms should be explained better in the paper with provision of relevant references and be careful on their numbering. I would like to see also some more particular and specific data regarding temperatures, precipitations, soil texture and other.
I suggest to remove word „soil” from the title, as the term „land” includes soil, but has much broader significance (soil+relief+climate, just in simplified way): „Identification of cultivated land soil quality grade fused with multis-source data and multi-temporal crop remote sensing information”.
Figures 2, 3, 5, 6 and 7 are difficult in interpretation at the zoom of 53%, which corresponds to A4 paper – I suggest to increase the whole figures at least 1,5 times, and the font 2-3 times.
DETAILED COMMENTS
Line 50: In my opinion, outside of overall land/soil quality, the authors should also check the possibility of quick identification of main limiting factors (e.g. water excess or deficiency, soil erosion etc) for crop production.
2. Materials and Methods
Line 99-102: Please, add more information regarding Shandong Province with respective references:
- area;
- altitude (minimum, maximum, average and median values);
- slope (minimum, maximum, average and median values);
- temperature and precipitation (minimum, maximum, average and median values, data from many years);
- designation of climate according to Koppen-Geiger classification (http://koeppen-geiger.vu-wien.ac.at/pdf/1901-1925.pdf or other sources) or other, commonly used in te world;
- designation of WRB (2015) Soil Reference Groups (https://www.isric.org/explore/wrb – check the links at the bottom of this page and consult soil scientist) prevailing in the province;
- specify prevailing soil texture classes according to WRB or USDA (they are almost identical)
Line 109: Please, specify, if „deep soil layer” refer sto it’s upper, usually dark and rather rich in humus, layer (A), which is most commonly known as „topsoil” or „accumulation layer”. The term „soil” refers not only to this upper layer, but includes also deeper layers (most frequently: eluviation and illuviation) which has been formed due to pedogenic processes.
Lines 114-115: Please, provide examples of crops, which are most frequently cultivated in two-crops season.
Lines 117-119: Please, recalculate and provide the power of machinery and amount of fertilizers used on hectare basis (i.e kW/ha and kgNPK/ha).
Before line 127: Please, create here new subchapter: "2.2. Soil sampling, analysis and data transformation" and ammend numeration of next subchapters
Line 131: The soil samples should be taken at least to the depth of about 90-100cm. The characterization of topsoil, at least in terms of soil texture is insufficient for the purposes of evaluation of land quality for majority of crops. This issue cannot be amended in this paper but consider it in future studies. Remember, that the soil with loamy sand layer in the top 30cm and underlaid by loam may be of better quality and productivity, than soil loam layer (0-30cm) underlaid by loamy sand.
Lines 133-134: Please, provide information of the number of samples taken (at least sum from the line 285)
Line 135: Please, add paragraph or a table on the definition and characterization of soil physical properties, which are mentioned in the Table 1 and respective references. It is hard to suppose, what is the exact definition and units (or other scale) for Irrigation and drainage capacity, Topography (slope, topographic wetness index or something else?), Cultivated layer texture (according to USDA or WRB - soil texture class or soil texture grouping, or rather according to Chinese classification, which, in such case, should be shortly described in supplementary file, if needed), texture configuration (?!), Salinization (EC or rather salt content, how it was measured?). Some of these indicators is mentioned in lines 143-145, but they should be defined precisely. I suggest to prepare a table with these characters and at least four columns: name (e.g. effective soil thickness), short definition, units, and source of definition.
Lines 136-141: Please add more detailed information regarding soil analysis methods (e.g. concentration of ammonium acetate for determination of potassium and extractant used before pH determination – was it distilled water, or 1N KCl, or other?) and respective references. Please add information and definitions and references for "Biodiversity" and "Farmland shelterbelt", which are mentioned in Table 1.
Line 146: Please, provide slightly more detailed information on IDW interpolation method, relevant reference and information, why this method was selected.
After line 148 an information regarding methodology of weights determination, which are provided in Table 1.
Line 149: I think, this is a title of subchapter and should be 2.2 which should be changed in2.3, if my suggestions are accepted.
Lines161 and others: Please add information of the producer (provider) and version of software used.
Line 206: Please, explain the P-S-R framework (pressure- state- response?)
Line 209: In my opinion, the term „limiting factor” would be more appropprate than „obstacle factor”.
Lines 217, 218, 219: please add the meaning for NDVI (Normalized Difference Vegetation Index?), DVI and RVI and sources, if others than [30] were used.
Lines 230-232: Please explain the indicators mentioned – ALI, AMI, AII and AFI. How are they calculated (or determined) and provide reference. Information on values of these indicators should be also provided, at least as supplementary file.
KMO and Bartlett test (line 364) were not mentioned in Materials and Methods, please, add some information and reference!
3. Results and analysis
The values of all indicators, obtained or calculated in the study and their descriptive statistics (minimum, maximum, median and mean values) should be provided in results or in supplementary file. I suggest to add some of them in Materials in Methods, as they contribute to characterization of area of study (altitude, slope, temperature, precipitations), however the interpretation of results of the study require knowledge of values of these indicators (soil quality indicators mentioned in Table 1, slope, precipitations, temperature, NDVI, DVI and RVI, ALI, AMI, AII and other mentioned in Table 2 and in the text).
Line 344-346: Here, the authors should cite particular information regarding slope, precipitation and temperature values, proving favourable natural conditions for crop growth
Figure 5: Besides increasing of the dimension of the figure and font, please, fit this figure and it's caption and not to a single page.
Table 3:
- please, fit the whole table 3 and the notes to a single page.
- please add information on meaning of all soil state indicators - x1-x8 - from formulas. Which indicator is designed as x1, x2 etc?
5. Conclusions
Lines 489-492: The conclusion (2) fits better to materials and methods (line 206) than to conclusions - it seems to be the supposition made at the beginning of the study.
Author Response

(The authors gave the same response as above.)

Reviewer 4 Report
Identification of Cultivated Land Soil Quality Grade Fused with Multi-source Data and Multi-temporal Crop Remote Sensing Information
Main comment:
The paper is in general interesting, since put together many different analysis techniques and information. On the other hand the approach is quite complicate. In particular the authors fail to clarify:
- how in pracatice the work can practically applied in land management
- a validation of the reported analysis is missing (in other word: how can you demonstrate that the final zoning you propose is correct?)
- also the materials and methods need to be bettere detailed.
Other comments to be addressed:
- Authors write that "plots near the sampling points were sampled along the "S" route,": please be more specific in the paper and write more extensively what you mean with "S" route
- Authors write that "At the same time, the project collected 1 : 500,000 soil map, geomorphic map, irrigation zoning map, and other basic maps.". What does it mean that the "project collected"? Report extensively in practice what was done.
- From Figure 1, apparently thousands of points were collected. How many soil samples weree collected? Please report.
- If properely interpreted the map, hundereds or thousands of soil samples have been collected. To collect so many samples I guess a quite long time is needed. Such time might introduce some bias or some drift (due to time variations in moisture or in sampling procedure by different operators,...). Pleasee discuss how you solved this issue.
- Sampling points in Figure 1 are much more than those in Figure 3. What is the relation between the two sampling points strategies?
- Please be more specific and (when not present) descibe in the paragraph the meaning of the quality parameters selectede and reported in Table 1. Also descibre ho te parameters have been in practice quantified/characterized (e.g. biodiversity: how did you measure it? the same for other parameters).
- To be considered homogeneous I believe a field has to have at least the samee crop, planted and harvested in the same dates. MODIS has a resolution of at least 250 m. So I wonder if it is reasonable to match and consider soil samples collected in a field with a size smaller than 250x250 m or 500x500 m. Did you consider a threeshold to avoid fields smaller than that? If not how can you couple large resolution satellitee data with small fields?
- Why not using Sentinel data, which might provide a higher resolution?
Minor comments to be addressed:
In general don't use "the" before "data"
In Figure 1 increase the size of the black spots indicating the sampling points (they are hardly visible)
Figure 3 is hardly readable. Please increease marker size or use multiple maps.
Figure 6: put the note not as a note but in the caption.
Please revise English (in particular some sentencees are quite long)
In case the paper will be accepted for revision, please address above comments and correct accordingly the paper,
- giving your pertinent comments in the “Response to reviewer” document
- reporting in the “Response to reviewer” document also the paragraph with amended text highlighted with yellow colour or the new amended figure.
Author Response

(The authors gave the same response as above.)

Round 2
Reviewer 1 Report
The authors have improved the overall quality of the manuscript as suggested and answered every question. I don't have any more comments. Only few minor checks shall be made by the authors as suggested below.
- L326: Here NDVI was introduced. It is too late. I suggest introduce it when it is first time used. Check for other such abbreviations throughout the manuscript.
- Table 1: Under name, you could simply use NDVI, DVI and RVI instead of full names.
- Table 2: Use ALI, AMI… so on under indicators instead of full names
Author Response

(The authors gave the same response as above.)

Reviewer 3 Report
The readability of this paper improved considerably. I still find some editorial and other issues, which should or might be corrected before publication...
I am thinking about a certain unification of the numbering of tables in supplementary materials. In the current version, there are four tables with number 1: in the main paper and in the supplementary materialas 1, 2, and 3. Please, consider some possibilities of amendment of this issue. For example, it is possible to create only one supplementary file and renumber the tables as ST1, ST2, ST3 and ST 4 (ST - Supplementary Table) or somehow similar. It is not obligatory, but it could additionally improve the lecture of the paper. If the authors accept this, please check al the places where these tables are mentioned.
The readability of the figures has improved, but Figures 5 and 6 are still not readable enough.
Consequently, I recommend to publish the paper after minor and rather small revision.
DETAILED COMMENTS
Lines 79-81: It is quite difficult to understand the second part of the phrase: "... the spectral indicators of crops, ..., can not only be used ...., but also be a good performance of soil quality".
Lines 121- 146: The authors provided all information I asked but not all sources! I would like to know (or find more easily):
- the sources regarding geographical and meteorological information;
- the classification of climate, according to which the designation DWA was used (I provided an example of source in my first review, suggesting Koppen-Geiger classification and a link as an example. If the authors decided to use this classification and source, they should provide respective information, or provide other name of classification and source which they used);
- according to other my suggestion authors provided names of WRB soil Reference Groups prevailing in the province, but additional information the WRB classification, it's version, as well as the respective, even very general chinese soil study should be referenced.
Line 154 and Table 2: The authors should clearly specify, if the amount (consumption)of chemical fertilizer 847kg/ha refers to sum of main macronutrients used (NPK), or rather to mass of commercial fertilizer. To be more clear, I provide an example: the notrogen (N) may be supplied as ammonium sulfate (22%N), ammonium nitrate (34%), urea (46%) and many other commercial, chemical fertilizer). However, the application of a dose of 100kgN/ha requires about 454,5kg/ha of ammonium sulfate, 303kg/ha of ammonium nitrate or 217kg/ha of urea. For this reason the amount of chemical fertilizers used, is frequently provided in statistics as the sum of so called pure nutrients of main elements (N, P and K) per hectare. I suppose the same is used in this study, but I would like to see it clearly conformed.
Line 177: I suppose that soil samples were taken from the soil layer between 0 and 20cm, and not exactly from the depth of 20cm. If I am right, I suggest to modify the part of this phrase to "... the sampling depth was up to 20cm", or similar and more precise way.
Line 181: "the soil samples ...were .... screened, ''? Please check, the phrase. Perhaps the authors mean "seeved"? If yes, please, provide mesh diameter (1 or 2mm? Or other?).
Table 2: Please, ammend the abbreviation of energy unit. I think it should be kW/ha, and not kw/ha.
Line 439-440: In my opinion, it would be better to replace "The spatial differentiation characteristics of soil types in different grades is obvious ..." with "The spatial distribution .of different soil quality grades is related mainly to topographic factors", after checking, if my suggestion reflects the authors' intention and is grammatically correct.
Tables 4 and Table 1 from supplementary material 3 (suggested ST4): The term "time phase" needs explanation! As I can suppose, it refers to the number of days of year. Similar, perhaps better explanation is needed in the text (current line 370?) and at footnotes of the tables.
Supplementary material 1, Table 1 (Suggested ST1): for chemical analyses (organic matter, available phosphorus, rapidly available potassium , total nitrogen and pH value (not PH value!), the authors should shift information on analysis method (e.g. for organic matter: "potassium dichromate - sulfuric acid") to the second column (Description) and quite extend it. E.g for organic matter the authors should inform that organic carbon (C) is measured, and for rapidly available potassium (we call it more frequently "exchangeable potassium" provide information about the concentration of ammonium acetate (1mol dm-3), and for pH provide the kind and concentration of solution (KCL, 1mol dm-3), in which this parameter was measured. These information was provided in cover letter, but reader may need them for better understanding of the paper and future discussions. In the column "Indicator source" of the same table, the authors may provide simply "laboratory analysis" and, optionally, the full name of laboratory.
Author Response

(The authors gave the same response as above.)

Reviewer 4 Report
Sound motivation have been provided for unclear points and the paper has been clearly improveed, and is now ready for publication
Author Response
Thank you for the time and effort spent in reviewing the manuscript. We acknowledge your comments and suggestions very much, which are valuable in improving the quality of our manuscript. Wish you a happy life and smooth scientific research.